# Particulate Matter-Induced Neurotoxicity: Unveiling the Role of NOX4-Mediated ROS Production and Mitochondrial Dysfunction in Neuronal Apoptosis

**DOI:** 10.3390/ijms25116116

**Published:** 2024-06-01

**Authors:** Ji-Hee Kim, Kyu-Hee Hwang, Seong-Heon Kim, Hi-Ju Kim, Jung-Min Kim, Mi-Young Lee, Seung-Kuy Cha, Jinhee Lee

**Affiliations:** 1Department of Occupational Therapy, Soonchunhyang University, Asan-si 31538, Republic of Korea; jhk1111@sch.ac.kr; 2Department of Physiology, Yonsei University Wonju College of Medicine, Wonju 26426, Republic of Korea; kyuhee@yonsei.ac.kr; 3Department of Global Medical Science, Yonsei University Wonju College of Medicine, Wonju 26426, Republic of Korea; 4Mitohormesis Research Center, Yonsei University Wonju College of Medicine, Wonju 26426, Republic of Korea; 5Department of Environmental and Energy Engineering, Yonsei University, Wonju 26493, Republic of Korea; seongheo@yonsei.ac.kr; 6Department of Psychiatry, Yonsei University Wonju College of Medicine, Wonju 26426, Republic of Korea; sallypopo@hanmail.net; 7Department of Medical Science, Soonchunhyang University, Asan-si 31538, Republic of Korea; jm52402@naver.com (J.-M.K.); miyoung@sch.ac.kr (M.-Y.L.); 8Department of Medical Biotechnology, Soonchunhyang University, Asan-si 31538, Republic of Korea

**Keywords:** air pollution, particulate matter (PM), NOX4, reactive oxygen species (ROS), voltage-dependent anion-selective channel 1 (VDAC1)

## Abstract

Urban air pollution, a significant environmental hazard, is linked to adverse health outcomes and increased mortality across various diseases. This study investigates the neurotoxic effects of particulate matter (PM), specifically PM2.5 and PM10, by examining their role in inducing oxidative stress and subsequent neuronal cell death. We highlight the novel finding that PM increases mitochondrial ROS production via stimulating NOX4 activity, not through its expression level in Neuro-2A cells. Additionally, PMs provoke ROS production via increasing the expression and activity of NOX2 in SH-SY5Y human neuroblastoma cells, implying differential regulation of NOX proteins. This increase in mitochondrial ROS triggers the opening of the mitochondrial permeability transition pore (mPTP), leading to apoptosis through key mediators, including caspase3, BAX, and Bcl2. Notably, the voltage-dependent anion-selective channel 1 (VDAC1) increases at 1 µg/mL of PM2.5, while PM10 triggers an increase from 10 µg/mL. At the same concentration (100 µg/mL), PM2.5 causes 1.4 times higher ROS production and 2.4 times higher NOX4 activity than PM10. The cytotoxic effects induced by PMs were alleviated by NOX inhibitors GKT137831 and Apocynin. In SH-SY5Y cells, both PM types increase ROS and NOX2 levels, leading to cell death, which Apocynin rescues. Variability in NADPH oxidase sources underscores the complexity of PM-induced neurotoxicity. Our findings highlight NOX4-driven ROS and mitochondrial dysfunction, suggesting a potential therapeutic approach for mitigating PM-induced neurotoxicity.

## 1. Introduction

Urban air pollution presents a significant environmental challenge associated with adverse health outcomes and increased disease mortality [1]. According to the World Health Organization (WHO), air pollution results in over 7 million premature deaths worldwide yearly [2,3]. The US Environmental Protection Agency (EPA) defines atmospheric particulate matter (PM), or particle pollution, as a mixture of solid particles and liquid droplets in the air. These particles are categorized by size and aerodynamic properties into coarse particles (PM10) with diameters between 2.5 and 10 μm, fine particles (PM2.5) smaller than 2.5 μm, and ultrafine particles (UFPM) less than 0.1 μm. Multiple studies have shown that PM impacts respiratory and cardiovascular health and crosses the blood–brain barrier (BBB) [4], directly affecting the brain. This leads to neurotoxic effects primarily via oxidative stress and subsequent neuronal cell death [5,6,7,8]. Furthermore, the size of PM particles correlates with their neurotoxic potential, indicating a size-dependent effect on neurotoxicity.

Despite the recognized link between air pollution and neurological damage, the precise mechanisms through which PM compromises neuronal integrity remain elusive. Notably, apoptosis and mitochondrial dysfunction are interconnected processes that can compromise the integrity and function of neurons. Ku et al. (2016) reported that exposure to PM2.5 impaired spatial learning and memory in mice, linked to brain mitochondrial injuries and altered apoptotic protein expression such as p53, bax, and Bcl2 [9]. Similarly, PM2.5 exposure in rats led to memory deficits due to brain oxidative stress and mitochondrial changes [10]. Furthermore, PM2.5-induced mitochondrial dysfunction and subsequent apoptosis can be mitigated by inhibiting VDAC1 in airway epithelial cells [11]. Although PM accelerates neural degeneration and increases the risk of mental disorders [12,13], the specific role of VDAC1 in neuron cell death due to PM exposure has not been thoroughly explored.

The cellular toxicity and apoptosis observed following PM exposure are primarily attributed to aberrant ROS production [14]. This PM-mediated ROS signaling can arise from various mechanisms, including inflammatory responses [15]. Oxidative stress, a well-established pathogenic factor, leads to neurotoxicity and neuronal cell death, contributing to neurodegenerative diseases such as Alzheimer’s disease, Parkinson’s disease, ischemic brain injury, stroke, and depression [6,16,17,18,19]. NOX4, uniquely located in the mitochondrial inner membrane, is vital in generating mitochondrial-derived ROS [20]. Recent studies have shown that PM2.5 exposure increases ROS generation by upregulating NOX4 expression in various tissues, including skin, lungs, heart, kidneys, and blood vessels [21,22,23,24,25]. However, the direct connection between NOX4 activation and PM-induced neurotoxicity is yet to be clarified.

This study aims to unravel the mechanisms by which PM2.5 and PM10 trigger neuronal apoptosis, focusing on the upregulation of VDAC1 and ROS generation through NOX4 activation. By comparing the effects of PM2.5 and PM10 on Neuro-2A cells, we aim to elucidate the reasons behind the increased susceptibility of cells to death following exposure to PM2.5. Our research reveals that PMs increase mitochondrial ROS production via stimulating NOX4 activity but not its expression level. Additionally, we evaluate a novel pathological role of PM linking VDAC1-mediated neuronal cell death and abnormal mitochondrial ROS generation driven by enhanced NOX4 activity in Neuro-2A cells.

## 2. Results

### 2.1. PM-Induced Neurotoxicity Causes Neuronal Cell Apoptosis

We investigated the differential neurotoxic effects of PM2.5 and PM10 on neuronal cells, focusing on cell survival and apoptosis. The MTT assay revealed a significant, dose-dependent decrease in cell viability upon exposure to both types, with PM2.5 demonstrating more significant toxicity than PM10 in Neuro-2A (Figure 1A, Appendix A) and SH-SY5Y cells (Appendix A). We then examined the mitochondrial-mediated apoptosis pathway, specifically analyzing VDAC1, cleaved-caspase3 (c-caspase3), Bax, and Bcl2. PM2.5 significantly upregulated VDAC1 in different concentrations, starting at 1 µg/mL in both cell lines, whereas PM10 required a higher dose of 10 µg/mL for a comparable effect in Neuro-2A cells and was less effective in SH-SY5Y cells (Figure 1B and Appendix A). Both PM types activated the c-caspase3 pathway, increasing pro-apoptotic BAX and decreasing anti-apoptotic Bcl2 in both cell lines (Figure 1B and Appendix A). These data suggest that PM2.5 and PM10 reduce cell viability by upregulating VDAC1 and activating the c-caspase3-mediated apoptosis pathway in neuronal cells.

### 2.2. PM Increases NOX4 Activity without Altering Protein and mRNA Expression in Neuro-2A Cells

PMs are known to produce ROS, leading to neurotoxicity. NOX proteins, particularly NOX4, serve as primary enzymatic sources of ROS in neurons during acute neuronal injuries and chronic neurodegenerative diseases [17,26]. RT-PCR analysis revealed NOX4 as the predominant isoform in Neuro-2A cells (Figure 2A). Despite varying PM doses, NOX4 mRNA and protein expression levels remained constant (Figure 2B–G), suggesting that PM does not influence the transcriptional regulation or protein stability of NOX4. However, total NOX activity showed a significant dose-dependent increase with both PM types, especially with PM2.5 exposure (Figure 2H,I). Conversely, in SH-SY5Y cells expressing multiple NOX proteins, PMs increased mRNA and protein expression of NOX2 but not NOX4 (Appendix A). This result indicates that PM2.5 and PM10 amplify ROS production by stimulating NOX activity rather than by altering NOX4 expression in Neuro-2A cells, unlike NOX2 expression.

### 2.3. PMs Facilitate Cytosolic and Mitochondrial ROS Generation in Neuro-2A Cells

PM interferes with mitochondrial electron transport chain (ETC) complexes I and III, enhancing the interaction of electrons with oxygen and subsequent ROS generation [27]. This effect on mitochondrial function is depicted in Figure 3A. Using DCF and MitoSOX, we quantified cytosolic and mitochondrial ROS production, respectively. PM2.5 significantly increased ROS levels in a dose-dependent manner, more so than PM10 (Figure 3B,C). Mitochondrial ROS, intensified by Antimycin A, an inhibitor of mitochondrial respiration, also rose more substantially with PM2.5 exposure in a dose-dependent manner (Figure 3D–H). These results indicate that both cytosolic and mitochondrial ROS are involved in PM-induced neurotoxicity, with PM2.5 exerting a more substantial effect than PM10 in Neuro-2A cells.

### 2.4. Inhibition of NOXs Mitigates PM-Induced Neurotoxicity and Neuronal Apoptosis

NOX2 and NOX4 are principal NOX isoforms in neuronal cells [28,29]. To further explore PM-induced neurotoxicity and apoptosis, Neuro-2A cells were treated with PM2.5 and PM10 along with Apocynin (Apo) and GKT137831 (GKT), inhibitors of NADPH oxidase and NOX1/4, respectively. Apo and GKT were used at 10 µM, approximately 10 times their EC50, as shown in Appendix A. Both inhibitors significantly reduced PM-derived and mitochondrial ROS production, implicating NOX4 in PM-triggered ROS production (Figure 4A–D). The inhibition of NOX activities by Apo and GKT attenuated the PM-induced increase in NOX activity (Figure 4E,F) and restored cell viability impaired by PM exposure (Figure 4G,H). Western blot analysis demonstrated that the increased BAX/Bcl2 ratio induced by PM2.5 and PM10 was reversed by the inhibitors (Figure 4G,H), supporting the hypothesis that PM2.5 and PM10 trigger neuronal apoptosis by promoting ROS generation through NOX4 activation in Neuro-2A cells. Meanwhile, both PM2.5 and PM10 promoted ROS production in SH-SY5Y cells expressing NOX2 (Appendix A). Unlike in NOX4-expressing Neuro-2A cells, PM-regulated apoptosis mediators and cell death in SH-SY5Y cells were rescued by Apo but not by GKT (Appendix A). Together, these data suggest that inhibition of NOX4 and NOX2 ameliorates PM-caused neurotoxicity and apoptosis.

## 3. Discussion

This study elucidates the novel pathological role of PM in exacerbating neuronal cell death, focusing on the upregulation of VDAC1 and subsequent mitochondria-derived ROS production that stimulates NOX4 activity. Our findings indicate that PM2.5 exerts a more pronounced cytotoxic effect than PM10. Both PM types induce cytosolic and mitochondrial ROS generation, upregulating NOX4 activity and leading to neuronal cell death. The neurotoxic effects, mediated by the BAX/Bcl2 apoptosis pathway and increased NOX4 activity and ROS production, were mitigated by NOX4 inhibition, highlighting a critical pathophysiological mechanism of PM-induced neuronal cell death via the NOX4–ROS–VDAC1–BAX/Bcl2-mediated apoptosis pathway.

Despite extensive studies into the mechanisms linking PM exposure to neuronal vulnerability and associated neurological disorders, definitive conclusions remain elusive. Li et al. (2020) provided evidence that PM2.5 can penetrate the blood–brain barrier (BBB) [4], causing neuronal loss and inflammation [30,31], thereby implicating fine PM in cognitive impairment and depressive symptoms via BBB penetration. Moreover, our results demonstrate that PM2.5 presents higher overall cytotoxicity (e.g., reduced cell viability, increased cytROS, mtROS, and NOX4 activity) than PM10. Due to their smaller size, PM2.5 particles can more readily penetrate cells via endocytosis, potentially leading to higher cytotoxicity [32]. Their smaller size facilitates rapid absorption into cells, triggering detrimental physiological responses such as increased cytosolic and mitochondrial ROS production and elevated NOX4 levels.

The brain’s susceptibility to oxidative stress is attributed to its high energy demands, limited endogenous antioxidants, dense neural networks, and high lipid and protein content [33]. NOX proteins, especially NOX4, serve as primary sources of ROS in the central nervous system (CNS) [34], often upregulated by stress and disease conditions like ischemic brain injury [26]. Interestingly, CNS-specific NOX2 deficiency in mice has been linked to impaired synaptic plasticity and memory formation [35], associating neurotoxicity with potential cognitive and depressive disorders [19,36]. Considering our Appendix A from SH-SY5Y cells, NOX2 has been identified as the primary source of ROS [28]. Despite the established role of NOX proteins in various neural damage and brain inflammation, the specific regulation of NOX proteins by PM in the brain remains underexplored. While previous studies have shown that PM2.5 can upregulate NOX4-induced ROS generation in cardiac tissues [22], our study reveals that PM2.5 and PM10 induce aberrant ROS generation by increasing NOX4 activity rather than upregulating its expression in Neuro-2A cells, offering new insights into PM-induced neurotoxicity.

PM-induced neurotoxicity also involves mitochondrial dysfunction, with PM2.5 and PM10 leading to mitochondrial permeability transition pore (mPTP) opening, membrane potential dissipation, respiratory disruption, and cell death [11,27,37]. Our findings of increased VDAC1 expression in response to PM exposure align with previous studies [11], suggesting that PM-induced mPTP opening contributes to excessive mitochondrial ROS production linked to lipid peroxidation and apoptosis. Bcl2 family proteins, including BAX and Bcl2, intricately control apoptosis by modulating VDAC-induced mitochondrial membrane permeability, releasing cytochrome c [38]. Furthermore, we observed BAX/Bcl2 ratio changes upon PM exposure, indicating a shift towards pro-apoptotic signaling. NADPH oxidase inhibitors (GKT137831 and Apocynin) effectively attenuated this shift, underscoring the role of the NOX4–ROS–VDAC1–BAX/Bcl2 pathway in PM-induced apoptosis. However, in SH-SY5Y cells, the increase in ROS induced by PMs was not suppressed by the NOX1/4 inhibitor, GKT137831, suggesting that the elevated NOX2 from PMs is considered the primary factor. Similarly, GKT137831 had minimal restorative effects on the BAX/Bcl2 ratio and did not rescue cell viability compared to Apocynin (Appendix A).

PM-induced neurotoxicity encompasses mechanisms beyond the NOX4-mediated apoptosis pathway, including endoplasmic reticulum stress, ERK/CREB pathway inhibition, ferroptosis induction [8,39], and inflammatory cytokine secretion [31]. PM-mediated ROS generation involves multiple pathways, such as the inflammatory response, hydroxyl radical formation, and NOX activation. Our findings, coupled with existing research [24], position NOX4 as a central mediator of PM-induced ROS production, particularly given its mitochondrial localization and potential role in ferroptosis, a process linked to mitochondrial metabolism impairment in Alzheimer’s disease [17].

While our study significantly advances our understanding of PM-induced neurotoxicity, it also underscores the need for further research. Specifically, future investigations should delve deeper into the complex mechanisms of PM-induced neurotoxicity, considering the influence of neighboring microglial cells [18] and astrocytes [16] on neuronal degeneration and overall brain function. Additionally, the in vitro nature of our study necessitates in vivo studies to validate these findings. Furthermore, the real-world variability in PM composition calls for more nuanced studies to assess the impact of PM exposure on human neural cells and brain function across diverse populations.

In conclusion, our study provides significant insights into the neurotoxic effects of PM, particularly emphasizing the critical role of NOX4-mediated ROS generation and VDAC1 upregulation in Neuro-2A. The heightened sensitivity of neuronal cells to PM2.5, compared to PM10, suggests that targeting NOXs could be a promising therapeutic strategy to mitigate PM-induced oxidative stress and neuronal damage. These results not only enhance our understanding of how elevated levels of fine PM may increase the risk of neurological disorders but also offer pontential evenues for therapeutic intervention.

## 4. Materials and Methods

### 4.1. Cell Culture

The Neuro-2A (CCL-131) murine neuroblastoma cell line and SH-SY5Y human neuroblastoma cell line were procured from the American Cell Type Culture Collection. Cells were cultured in Eagle’s Minimum Essential Medium (EMEM) (#30-2003, ATCC, Manassas, VA, USA) supplemented with 10% fetal bovine serum (#16000-044, Gibco, Grand Island, NY, USA) and 1% penicillin/streptomycin (#SV30010, HyClone, Logan, UT, USA) in a humidified incubator at 37 °C and 5% CO_2_.

### 4.2. Materials

PM2.5 (SRM 2786, Fine Atmospheric Particulate Matter < 4 μm (mean, 2.8 μm), NIST, Gaithersburg, MD, USA) [40] and PM10 (ERM-CZ100, #0647, JRC, Geel, Belgium) [41] were dissolved in phosphate-buffered saline (PBS). GKT137831 (#17764, Cayman Chemical Company, Ann Arbor, MI, USA), Apocynin (#178385, Sigma-Aldrich), and Antimycin A (#A8674, Sigma-Aldrich) were dissolved in dimethyl sulfoxide (DMSO).

### 4.3. MTT Assay

Neuro-2A and SH-SY5Y cells were plated at 1 × 10^4^ cells/well in a 96-well plate. GKT137831 and Apocynin were prepared in DMSO as the solvent and pretreated to cells for 1 h before a 24 h incubation with PM2.5 and PM10 in complete media. PBS or DMSO (final concentration is 0.1%) were used as a vehicle. The MTT solution was prepared by dissolving 3-[4,5-dimethylthiazol-2-yl]-2,5-diphenyl tetrazolium bromide (#M2128, Sigma-Aldrich) in fresh EMEM at 0.5 mg/mL. After treatment, cells were incubated with 100 μL of MTT solution in an incubator for 2 h. After removing the medium, DMSO was added to each well to dissolve formazan, and samples were gently shaken for 10 min. The plates were read using an EpochTM Microplate Spectrophotometer (Bio-Tek, Winooski, VT, USA) at 570 nm with a reference wavelength of 650 nm.

### 4.4. Measurement of Reactive Oxygen Species (ROS)

Cytosolic ROS was measured as described previously [42]. GKT137831 (5 μM) and Apocynin (10 μM) were pretreated for 10 min before a 1 h incubation with fine dust (PM2.5 or PM10). Neuro-2A and SH-SY5Y cells were loaded with 5 μM CM-H_2_DCF-DA (5-(and-6)-chloromethyl-20,70-dichlorodihydrofluorescein diacetate, acetyl ester, #D399, Invitrogen) for 20 min at 37 °C. Fluorescence intensities were monitored in a multi-well fluorescence reader (FlexStation, Molecular Devices).

Cells were loaded with 5 μM MitoSOX (#M36008, Molecular Probes, Invitrogen) for 10 min in warmed PBS at 37 °C and washed with warm PBS [42]. GKT137831 (5 μM) and Apocynin (10 μM) were incubated for 10 min before a 1 h co-incubation with fine dust (PM2.5 or PM10; 100 μg/mL). The complex III inhibitor, Antimycin A (5 μM for 30 min), was used as a positive control. Images of MitoSOX fluorescence were collected within 5 min using 510 nm for excitation and 580 nm emission (IX-81; Olympus, Tokyo, Japan) with a confocal spinning disk (CSU10; Yokogawa, Tokyo, Japan). Data from more than ten images from three independent experiments were averaged after background fluorescence correction using MetaMorph 6.3 software (Molecular Devices).

### 4.5. NOX Activity Assay

Total NOX activity was measured as described previously [43] with minor modification. GKT137831 (5 μM) and Apocynin (10 μM) were applied for 1 h before a 24 h co-incubation with PM2.5 or PM10 (100 μg/mL, each). Neuro-2A cells were washed twice with warm PBS and scraped from the dishes with PBS containing protease inhibitor; samples were centrifuged at 2000 rpm for 3 min at 4 °C. Cell pellets were dissolved in lysis buffer (20 mM KH_2_PO_4_, 1 mM EGTA, protease inhibitor cocktail, pH 7.0). The cell suspensions were homogenized with 100 strokes in a Dounce homogenizer, and the homogenates (50 μg) were added to 50 mM phosphate buffer (pH 7.0), including 1 mM EGTA, 150 mM sucrose, 5 μM lucigenin, and 100 μM NADPH. The reaction generated photon particles that were detected by a luminometer (Synergy 2, BioTek Instruments, Winooski, VT, USA) at every 20 s for 10 min, and the results were expressed as relative light units (RLU). The result of the blank buffer was subtracted from each test reading, and superoxide generation was expressed as RLU/min/μg protein.

### 4.6. RT- and Real-Time qPCR

Total RNA was extracted and purified from trypsinized Neuro-2A cell pellets using a Hybrid-RTM total RNA purification kit (#305-101, GeneAll, Seoul, Republic of Korea) according to the manufacturer’s instructions. Complementary DNA (cDNA) was synthesized from 1 μg of total RNA using a ReverTraAce^®^ qPCR RT Master Mix with gDNA Remover (#FSQ-301, Toyobo, Osaka, Japan). The mRNA abundance was analyzed by real-time quantitative PCR with SYBR Green (#204143, Qiagen, Germantown, MD, USA). Primer sequences are in Appendix A. The experiments were performed in triplicate in a real-time PCR instrument (7900HT, Thermo Fisher Scientific). Gene expression was determined using the 2^−ΔΔCt^ method with 18s as the reference gene.

### 4.7. Western Blot

Western blotting (WB) was performed as described previously [44]. Briefly, cells were mechanically homogenized in RIPA lysis buffer (#89900, Thermo Fisher Scientific Inc., Waltham, MA, USA) containing protease inhibitor cocktail (#P8340, Sigma Aldrich, St. Louis, USA) and phosphatase inhibitor PhosStop (#4906837001, Roche, Basel, Switzerland). The following primary antibodies were used for immunoblotting: NOX4 (#ab133303, diluted 1:1000) and β-actin (#ab6276, diluted 1:10,000) were purchased from Abcam (Cambridge, UK) and VDAC1 (#4866, diluted 1:1000), BAX (#2772, diluted 1:1000), Bcl2 (#3498, diluted 1:1000), and cleavage-caspase3 (#9661, diluted 1:1000) were obtained from Cell Signaling Technology Inc. (Beverly, MA, USA). GAPDH was purchased from Santa Cruz Inc. (sc-365062, diluted 1:3000). Bands were detected and quantified using the ChemiDoc XRS+ Imaging System with Image Lab Software (version 5.2.1 build 11, BioRad, CA, USA) or Vilber Lourmat (Fusion Solo 6S Edge V.070, France) with its analysis software.

### 4.8. Statistical Analysis

The Student’s *t*-test was used to compare the means of the two groups. One-way ANOVA was used to compare the means of three or more groups. All graphs were generated, and statistical analysis was performed using Prism version 10.0, GraphPad Software. Data are presented as mean ± SEM, and *p* ≤ 0.01 was considered significant.

## Figures and Tables

**Figure 1 ijms-25-06116-f001:**
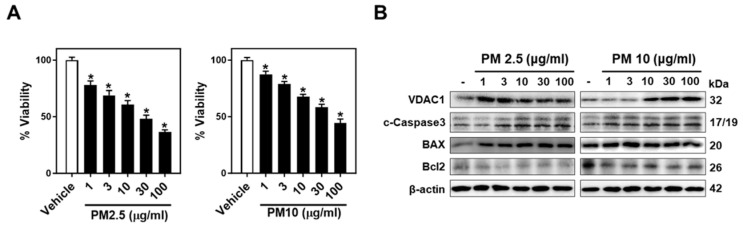
PM2.5 and PM10 induce cell death through mitochondria-mediated apoptosis in Neuro-2A cells. (**A**) Effect of PM2.5 (left) and PM10 (right) on the cell viability of Neuro-2A cells assessed by the MTT assay. Cells were incubated with different concentrations (1, 3, 10, 30, and 100 μg/mL for 24 h) and PBS was used as the control (vehicle). (**B**) Representative WB showing the dose-dependent effects of PM2.5 (left) and PM10 (right) treatment (1, 3, 10, 30, 100 μg/mL for 24 h) on VDAC1, cleaved (c)-caspase3, BAX, and Bcl2 in Neuro-2A cells. β-actin was used as a loading control. Data were presented as mean ± SEM, * *p* ≤ 0.01 vs. Vehicle (PBS). Data were analyzed by one-way ANOVA in (**A**).

**Figure 2 ijms-25-06116-f002:**
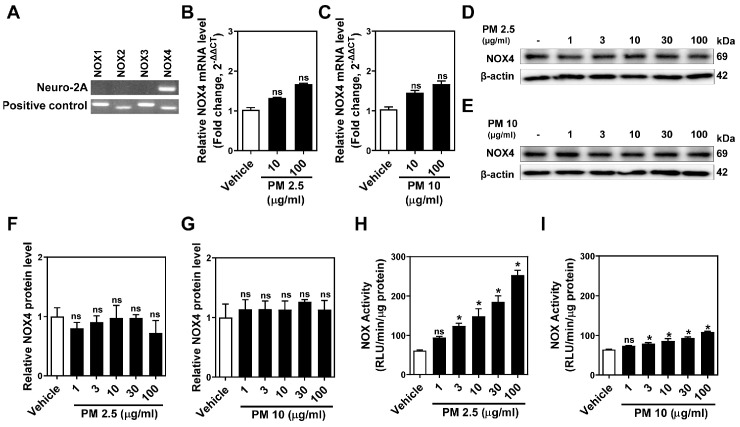
PM2.5 and PM10 increase NOX activities without altering mRNA and protein expression levels in Neuro-2A cells. (**A**) RT-PCR analysis showing mRNA levels of NOX1-4 in Neuro-2A cells. cDNAs for positive control were used: mouse kidney for NOX1, 3, and 4 and mouse liver for NOX2. (**B**,**C**) Real-time qPCR for NOX4 mRNA in response to different concentrations (10, 100 μg/mL) of (**B**) PM2.5 and (**C**) PM10 for 24 h. PBS was used as vehicle control. (**D**,**E**) Representative immunoblotting showing NOX4 expression after treatment with different doses of (**D**) PM2.5 and (**E**) PM10 (1, 3, 10, 30, 100 μg/mL) for 24 h. β-actin was used as loading control. Blots were repeated three times independently. (**F**,**G**) Quantification of NOX4 levels from all three WB. (**H**,**I**) Total NOX activity was measured in Neuro-2A cells with (**H**) PM2.5 and (**I**) PM10 stimulation (1, 3, 10, 30, 100 μg/mL, for 1 h). All values are mean ± SEM. * *p* ≤ 0.01 vs. Vehicle; ns, not significant. Data were analyzed by one-way ANOVA in (**B**,**C**) and (**F**–**I**).

**Figure 3 ijms-25-06116-f003:**
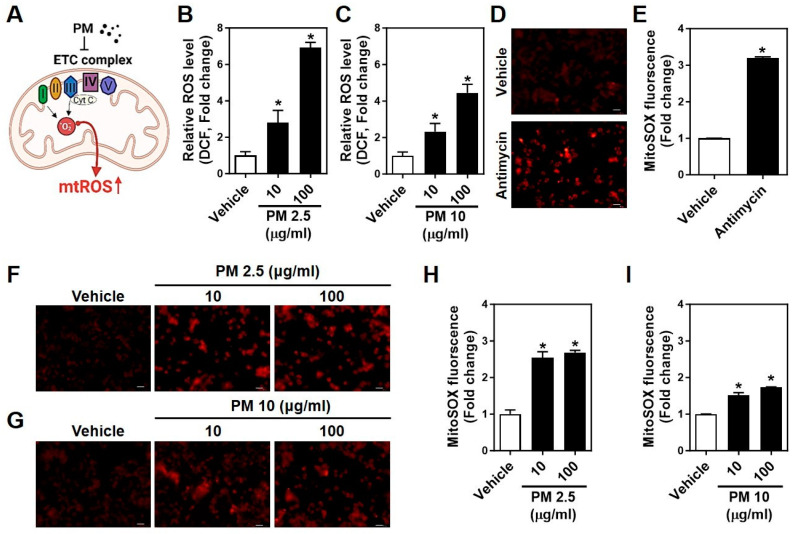
PM2.5 and PM10 increase cytosolic and mitochondrial ROS production in Neuro-2A cells. (**A**) The illustration represents the disruption of the electron transport chain (ETC) by PM infiltration into the mitochondria, boosting ROS production (mtROS). (**B**,**C**) ROS measurement using DCF after treatment with different doses (10, 100 μg/mL for 1 h, each) of (**B**) PM2.5 and (**C**) PM10 in Neuro-2A cells. Cytosolic ROS (ΔDCF intensity) was analyzed after subtraction of the basal level (0 h). (**D**) Representative confocal images of intracellular mitoSOX shows Antimycin A (5 μM for 30 min) induction of mitochondrial superoxide in Neuro-2A cells. (**E**) Quantitation of mitoSOX fluorescence intensity in panel D. (**F**,**G**) Representative confocal images of intracellular mitoSOX by (**F**) PM2.5 and (**G**) PM10 (10, 100 µg/mL for 1 h, each) in Neuro-2A cells. (**H**,**I**) Quantitation of mitoSOX fluorescence intensity in panels F and G, respectively. Scale bars, 5 μm. All values are the mean ± SEM. * *p* ≤ 0.01 vs. Vehicle. One-way ANOVA in (**B**,**C**) and (**H**,**I**); Unpaired Student’s *t*-test for (**E**).

**Figure 4 ijms-25-06116-f004:**
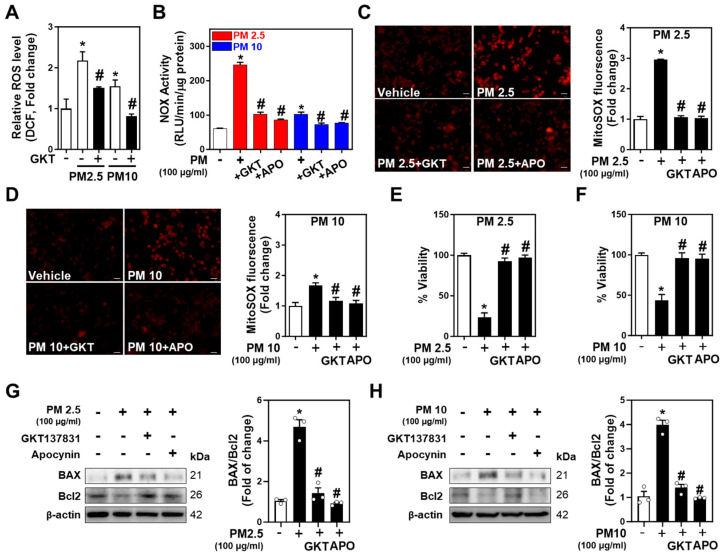
NADPH oxidase inhibitors rescue PM-induced cell viability reduction in Neuro-2A cells. (**A**) GKT137831 (GKT, 10 μM for 10 min) suppressed PM2.5- and PM10-mediated (100 μg/mL for 1 h) cytosolic ROS (ΔDCF intensity) generation in Neuro-2A cells. (**B**) Inhibition of NOX4 activity by GKT (10 μM for 10 min) or inhibition of NADPH oxidase activity by APO (10 μM for 10 min) alleviated PM2.5- and PM10-induced (100 μg/mL for 1 h) total NOX activity in Neuro-2A cells. (**C**,**D**) Representative confocal images of intracellular mitoSOX incubated with (**C**) GKT137831 (GKT, 10 μM) and (**D**) Apocynin (APO, 10 μM) pretreated for 10 min before 1 h of co-incubation with fine dust (PM2.5 or PM10; 100 μg/mL) and quantitation of mitoSOX fluorescence intensity. Scale bars, 5 μm. (**E**,**F**) Reduction of cell viability by (**E**) PM2.5 and (**F**) PM10 was recovered by treatment with GKT137831 (10 μM) and Apocynin (10 μM) in Neuro-2A cells. The inhibitors were pretreated for 30 min before 24 h co-incubation with PM2.5 or PM10. (**G**,**H**) The protein expressions of BAX and Bcl2 were detected in Neuro-2A cells, treated or not with GKT137831 (10 μM) and Apocynin (10 μM), in the presence or absence of (**G**) PM2.5 and (**H**) PM10 for 24 h. Quantification of Bax/Bcl2 levels normalized with β-actin, respectively. Blots were repeated three times independently. All values are mean ± SEM, * *p* ≤ 0.01 vs. Vehicle (0.1% DMSO, *v*/*v*), # *p* ≤ 0.01 vs. PM2.5 or PM10; one-way ANOVA was used for (**A**–**H**).

## Data Availability

The data used and/or analyzed during the current study are available from the corresponding author upon reasonable request.

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
