# Peer review of "Particulate Matter-Induced Neurotoxicity: Unveiling the Role of NOX4-Mediated ROS Production and Mitochondrial Dysfunction in Neuronal Apoptosis"

_ijms, 2024, doi:10.3390/ijms25116116_

Round 1

Reviewer 1 Report (New Reviewer)

Comments and Suggestions for Authors

In the study titled “Particulate Matter-Induced Neurotoxicity: Unveiling the Role of NOX4-Mediated ROS Production and Mitochondrial Dysfunction in Neuronal Apoptosis,” the authors investigate the neurotoxic effects of particulate matter, focusing on their role in triggering oxidative stress and subsequent neuronal cell death. The manuscript is well-structured, containing necessary data, and aligns with the scope of this Journal. The introduction and background are appropriate considering the paper’s objectives. The figures and tables are comprehensive and informative. However, the manuscript requires some corrections before it can be published. My comments on this article are as follows:

1.    The manuscript contains some grammatical errors. It is strongly recommended that the authors revise the paper for grammar.

2.    It would be beneficial to include some numerical values from the results in the abstract.

3.    The authors should consider adding a paragraph detailing the annual global death statistics due to air pollution.

4.    The novelty and contributions of this work, compared to other studies in the literature, should be emphasized in the last paragraph of the introduction and in the Abstract section.

5.    Could the authors clarify the vehicle used for the cell viability assay? Was it DMSO or H2O? If DMSO was used, what percentage was utilized?

6.    In the MTT assay conducted in this study, which normal cells were used for comparison with the cancer cells?

7.    The authors have examined some key genes involved in apoptosis, such as caspase3, BAX, and Bcl2. However, they did not include the p53 gene family. Could the authors explain the reason for this?

Comments on the Quality of English Language

 Minor editing of English language required

Author Response

Comments 1: The manuscript contains some grammatical errors. It is strongly recommended that the authors revise the paper for grammar.

Response 1: Thank for pointing this out. To improve the quality of the paper, we have had the manuscript edited by the highly qualified native English-speaking editor. The revised manuscript has been edited to fit proper English language, grammar, punctuation, spelling, and overall style.

Comments 2: It would be beneficial to include some numerical values from the results in the abstract.

Response 2: Thank you for your comment. To clarity our results, we have added numerical values for PM concentrations and the fold increase in their effects on ROS production and NOX4 activity.

Comments 3: The authors should consider adding a paragraph detailing the annual global death statistics due to air pollution.

Response 3: Following your suggestion, we added the annual global death statistics due to air pollution and references to the revised manuscript (line 49-51).

Comments 4: The novelty and contributions of this work, compared to other studies in the literature, should be emphasized in the last paragraph of the introduction and in the Abstract section.

Response 4: We thank valuable feedback. We appreciate the suggestion to highlight the novelty and contributions of our work more prominently. Our study provides unique insight into the differential regulation of NOX proteins by PMs, especially PM2.5 and PM10, in neuronal cells. We discovered that PMs increase mitochondrial ROS production by stimulating NOX4 activity rather than its expression. Additionally, we found that PMs induce ROS generation via increased expression and activity of NOX2 in SH-SY5Y cells. These findings suggest that PMs regulate NOX proteins differently, offering new perspectives on the mechanisms of PM-induced neurotoxicity. We emphasized these points in the last paragraph of the introduction and the Abstract section.

Comments 5: Could the authors clarify the vehicle used for the cell viability assay? Was it DMSO or H2O? If DMSO was used, what percentage was utilized?

Response 5: We appreciate the reviewer's inquiry regarding the vehicle used for the cell viability assay. In Figure 1A, we used PBS as the vehicle, which is described in the figure and its legend (line 101). We apologize for lacking information about the vehicle used in Figures 4E and 4F. For these figures, we used DMSO at a final concentration of 0.1% (v/v) as the vehicle to minimize potential cytotoxic effects. Apocynin and GKT137831 were prepared in DMSO as the solvent. We have added this information in lines 198 and 291-3.

Comments 6: In the MTT assay conducted in this study, which normal cells were used for comparison with the cancer cells?

Response 6: We appreciate insightful comment regarding comparing normal cells with cancer cells in the MTT assay. We agree that it is essential to compare the toxic effects of PM on normal and cancer cells, given that the cell lines used in this study, Neuro2-A and SH-SY5Y, are neuroblastoma cells rather than normal neurons. This could be a critical limitation for our study investigating PM-induced neurotoxicity. However, these cell lines are not derived from normal neurons and share some properties with cancer cells, they are valuable models for studying neurotoxicity. Neuro2-A and SH-SY5Y cells, immortalized neuroblastoma cells, were chosen for this study because they exhibit neuronal characteristics and are widely used in neurobiological research. Our research aimed to explore the mechanisms underlying mitochondrial dysfunction mediated by NOX4 in response to PM exposure, specifically in Neuro-2A cells. Although these cells are not perfect representations of normal neuronal cells, they are well-established in investigating neurotoxic effects. We recognize the value of your suggestion and will consider including primary cell comparisons in future studies to provide a more comprehensive understanding of the toxic effects of air pollution.

Comments 7: The authors have examined some key genes involved in apoptosis, such as caspase3, BAX, and Bcl2. However, they did not include the p53 gene family. Could the authors explain the reason for this?

Response 7: We appreciate the reviewer's observation regarding the absence of the p53 gene family and acknowledge that including it could provide valuable insights into apoptotic pathways. p53 is well-known as a tumor suppressor and has various functions beyond apoptosis, including inducing cell cycle arrest to allow for either repair and survival of the cell or apoptosis to eliminate the damaged cell (Front Aging Neurosci. 2022;14:835288). It has been reported that p53 modulates VDAC1 oligomerization, promoting the formation of high molecular mass complexes. These complexes play a critical role in the process of mitochondrial outer membrane permeabilization (MOMP), which is a critical event in the release of apoptogenic factors like cytochrome c from the mitochondria into the cytosol, leading to apoptosis (J Biol Chem. 2015;290(39):23563-78). However, no studies have examined the effects of PM on p53 in Neuro-2A cells. Therefore, future experiments investigating the role of p53 would be beneficial. In this study, our primary objective was to investigate PM-induced neurotoxicity, specifically examining the effects on cell viability caused by oxidative stress-mediated through VDAC1 and NOX4. As mentioned in the manuscript, we aimed to explore mitochondria-mediated apoptosis pathways focusing on mitochondrial ROS production via NOX and VDAC1-dependent cell death. Thus we did not focus on p53. We agree that the p53 gene family plays a critical role in apoptosis and affects other essential cellular functions, such as the cell cycle or regulation of VDAC1 oligomerization, which we did not examine in this study. We appreciate your valuable suggestion and will consider investigating the role of p53 in future studies to provide a more comprehensive understanding of apoptotic mechanisms in response to air pollution exposure.

4. Response to Comments on the Quality of English Language

Our response: We have carefully edited the manuscript to enhance its quality in terms of English language, grammar, punctuation, spelling, and overall style.

5. Additional clarifications

Our response: We carefully revised according to all reviewers’ comments. After revising according to the reviewers’ comments, the quality of this manuscript was definitely improved. We would like to express our deep appreciation for considering our manuscript in this journal.

Reviewer 2 Report (New Reviewer)

Comments and Suggestions for Authors

I appreciate the opportunity to review the manuscript. 

However, the authors should take note of the following major and minor remarks to improve the manuscript:

1. The authors have chosen two different types of particulate matter (PM). Most of the references cited have established PM2.5-induced neurodegeneration and related pathogenesis. It would be beneficial for the authors to provide a clear rationale behind the inclusion of PM10 in this experimental setup.

2. Additionally, it would be insightful to understand the reasoning for using different concentrations of PM2.5 and PM10 instead of optimizing the dose before conducting further experiments.

3. Despite the lack of significant differences in the relative mRNA and protein levels of NOX4 in Figure 2, the authors claim a direct involvement of NOX4-dependent ROS generation. While there are observable changes in NOX2 levels in the SH-SY5Y experiments (as seen in the supplementary data), the correlation to NOX4 remains unclear. The increased ROS could be an indirect correlation but the direct NOX4 activity measurement using the Amplex red assay could be more reliable, in my opinion. 

4. It is also unclear why the authors measured total NOX activity instead of specifically measuring NOX4. This choice needs further justification.

These revisions will help clarify the study's methodology and strengthen the interpretation of the results.

Comments on the Quality of English Language

At some point, there are grammatical mistakes. e.g., line no 151. 

Author Response

Comments 1: The authors have chosen two different types of particulate matter (PM). Most of the references cited have established PM2.5-induced neurodegeneration and related pathogenesis. It would be beneficial for the authors to provide a clear rationale behind the inclusion of PM10 in this experimental setup.

Response 1: Yes, most of the references cited have established PM2.5-induced neurodegeneration and related pathogenesis. However, PM10 can also pass through the blood-brain barrier, and long-term exposure to PM10 is related to neurodegenerative diseases such as Alzheimer's disease (J Alzheimers Dis. 2020;78(2):745-756). Similar to our results, it has been reported that PM2.5 exhibits more significant cytotoxicity than PM10 (Mutat Res. 2000;471(1-2):45-55).

While PM2.5 and PM10 are often mixed in the real world, distinct characteristics can exist in specific areas. The presence ratio of PM2.5 and PM10 varies depending on factors such as local levels of air pollution, geographical characteristics, climate conditions, and industrial and transportation activities. Typically, urban and highly industrialized areas exhibit higher concentrations of PM2.5 due to factors like traffic, industrial activities, and fuel combustion. Conversely, PM10, comprising larger particles, can originate from construction activities and road maintenance (Int J Environ Res Public Health. 2020 Sep; 17(18): 6585). Considering the regional variability in the ratio of PM2.5 to PM10, our research design incorporated both types of particulate matter.

This study will offer valuable insights into understanding the cytotoxicity of both PM2.5 and PM10, providing fundamental data for investigating the relationship between regional air pollution levels and disease incidence. In this study, cell viability, ROS production (measured by DCF and MitoSOX), and NOX4 activities were more sensitively affected by PM2.5. These results will provide crucial information for developing strategies to protect against PM-induced cytotoxicity.

Comments 2: Additionally, it would be insightful to understand the reasoning for using different concentrations of PM2.5 and PM10 instead of optimizing the dose before conducting further experiments.

Response 2: Thank you for your constructive comment. To optimize the dose for our experiments, we explored various concentrations of PMs ranging from 1 to 100 µg/ml. Our findings revealed differential effects on cell viability across different concentrations and cell types. Specifically, in Neuro-2A cells, both PM2.5 and PM10 exhibited decreased viability at 1 µg/ml, whereas significant viability reduction in SH-SY5Y cells was observed starting from 10 µg/ml for PM10. These points highlight the variability in cellular sensitivity to PM exposure.

Considering PM2.5 concentrations in the blood typically fall within the ng/ml range, other groups initially explored lower concentrations, such as 0.1 µg/ml or 0.01 µg/ml. However, these concentrations failed to elicit sufficient cellular responses within 24 hours, with a noticeable viability decrease only evident at a minimum concentration of 1 µg/ml, as supported by previous studies (Oncol Lett. 2018;16(2):2732-2740; Sci Prog. 2022;105(3):368504221113709). Consequently, concentrations below 1 µg/ml were not investigated for both PMs in this study.

Further analysis of VDAC1 and apoptosis pathway proteins indicated consistent alterations at concentrations of 10 µg/ml and above. While NOX4 protein and mRNA expression levels remained unaffected, NOX4 activity notably increased from 3 µg/ml for both PM2.5 and PM10. Additionally, cytosolic ROS measured by DCF did not exhibit a significant increase at 1 µg/ml, thus prompting the exclusion of this data point from Figure 2.

Subsequently, we determined the IC50 of PM-induced cell viability from Figure 1A, which was found to be 25 µg/ml for PM2.5 and 32.3 µg/ml for PM10 (Supplementary Figure 2A and B). Consequently, to elucidate the mechanism of ROS increase induced by PM in neural cells, we selected a concentration of 100 µg/ml, which reliably induced a cellular response, for assessing the inhibitory effect of NOX inhibitors.

The concentration of Apocynin and GKT137831 was determined based on their respective EC50 values (Supplementary Figure 2C and D). Viability measurements with different concentrations of both inhibitors revealed that 10 µM effectively counteracted the PM-induced decrease in viability. This additional data has been included in supplementary Figure 2 in this revised study version.

Comments 3: Despite the lack of significant differences in the relative mRNA and protein levels of NOX4 in Figure 2, the authors claim a direct involvement of NOX4-dependent ROS generation. While there are observable changes in NOX2 levels in the SH-SY5Y experiments (as seen in the Supplementary data), the correlation to NOX4 remains unclear. The increased ROS could be an indirect correlation but the direct NOX4 activity measurement using the Amplex red assay could be more reliable, in my opinion.

Response 3: We appreciate your insightful suggestion regarding using the Amplex Red assay for a more specific measurement of NOX4 activity. The Amplex Red assay could provide a more precise measurement of NOX4-specific activity, particularly in cell lines where multiple NOX isoforms are expressed.

In our study, we explored ROS generation by using DCF in both Neuro-2A and SH-SY5Y cell lines. As shown in Figure 3A or Supplementary Figure 3A, it is essential to note that Neuro-2A cells exclusively express NOX4, while SH-SY5Y cells predominantly express NOX2, although NOX4 is also present.

It is well established that NOX4 is the dominant isoform in Neuro-2A in other researcher groups (PLoS One. 2013;8(3):e58339.). Therefore, the ROS activity measured in Neuro-2A cells can be confidently attributed to NOX4. This assumption is supported by the high expression of NOX4 relative to other NOX isoforms, making Neuro-2A a suitable model for studying NOX4-related ROS production.

In SH-SY5Y cells, while NOX2 is the dominant isoform, expression of NOX3 and NOX4 is also observed. Although the Amplex Red assay would be appropriate for distinguishing NOX4-specific ROS production, specific NOX inhibitors can also help determine the source of ROS generation. In Supplementary Figure 3E, ROS generation induced by PM was not inhibited by GKT137831, a NOX1/4 inhibitor, indicating that NOX1 or NOX4 is unlikely to be responsible for the observed ROS increase. In Supplementary Figures 3H and I, the MTT assay showed that GKT137831 did not rescue the PM-induced reduction in cell viability, but Apocynin, a non-specific NOX inhibitor, protected it. This suggests that in SH-SY5Y cells, the ROS generation and subsequent apoptosis are primarily driven by NOX isoforms other than NOX1 or NOX4.

In conclusion, while our current data suggest that NOX4 does not play a significant role in ROS generation in SH-SY5Y cells, using the Amplex Red assay, as suggested, could provide further clarity and confirm these findings. We acknowledge the potential value of this assay and appreciate the reviewer’s valuable recommendation.

Comments 4: It is also unclear why the authors measured total NOX activity instead of specifically measuring NOX4. This choice needs further justification.

Response4: We agree with the reviewer’s point. Although we did not specifically measure NOX4 activity, it is important to note that in Neuro-2A cells which exclusively express NOX4 (Figure 3A). Therefore, the NOX activity measured in these cells can be primarily attributed to NOX4.

4. Response to Comments on the Quality of English Language

Our response: We have carefully edited the manuscript to enhance its quality in terms of English language, grammar, punctuation, spelling, and overall style.

5. Additional clarifications

Our response: We carefully revised it according to all reviewers’ comments. After revising according to the reviewers’ comments, the quality of this manuscript was improved. We would like to express our deep appreciation for considering our manuscript in this journal.

Round 2

Reviewer 1 Report (New Reviewer)

Comments and Suggestions for Authors

Accept, the authors have done a good work

Reviewer 2 Report (New Reviewer)

Comments and Suggestions for Authors

Thank you for making all the necessary changes. 

Regards,

Namdev

This manuscript is a resubmission of an earlier submission. The following is a list of the peer review reports and author responses from that submission.

Round 1

Reviewer 1 Report

Comments and Suggestions for Authors

The article by Kim et al. entitled ,, Particulate Matter-Induced Neurotoxicity: Unveiling the Role of NOX4-Mediated ROS Production and Mitochondrial Dysfunction in Neuronal Apoptosis’’ investigates the neurotoxic effects of particulate matters (PM) of 2.5 and 10 μm, by examining their role in inducing oxidative stress and subsequent neuronal cell death. Overall, I think this is an interesting and relatively novel concept and Authors made an effort to conduct this research. However, I have certain concerns regarding this work.

My detailed comments are given below:

Major points:

1.         The major study limitation is connected to the fact that it has been performed on a single cell line (Neuro-2A), which makes the conclusions specific to the tested cells. In certain journals it would be impossible to publish the results coming from a single cell line.

2.         Authors strongly rely on the results of Western blot analyses, which they show in a form of representative images and statistically analyzed graphs. This basically means that blots were performed as at least three independent experiments. In this respect, all membranes should be shown in the supplementary file with an indication of blots used in figures. Otherwise, it is unclear how blots were analyzed. If Authors are not able to provide three repeats of each membrane, Westerns should be presented without densitometric graphs and statistical significance markings, purely as a qualitative and not quantitative method.

Minor points:

1.         I suggest grouping the figure panels in frames, since having the numbering from A-L in one Figure is very hard to follow

2.        It seems that PM2.5 present higher overall cytotoxicity (e.g. higher ROS, MitoSox, NOX4 values) than PM10. Authors should discuss potential reasons.

3.         Editing error: line 82 should be ‘cells’, instead of ‘cell s’

Author Response

Thank you for your consideration of our manuscript and helpful guidance on improving its presentation. We have revised the manuscript according to with your guidance and the reviewers’ comments. Details of our revisions are described in our responses to the reviewers’ comments below.

The current revised manuscript contains additional results from experiments suggested by the reviewers, featuring experimental data from the additional neuronal cell line SH-SY5Y cells. We believe these addition results strengthen the reliability of our hypothesis and extend the applicability of our conclusions. In the re-submitted files, all changes have been highlighted and the revised texts are in red font to facilitate the review.

Sincerely yours,

Reviewer 2 Report

Comments and Suggestions for Authors

Given the growing health concerns of air pollution, this study is a topic of interest, and could provide valuable information for scientists and policy-makers. The authors demonstrated that PM exposure, especially PM2.5, causes neurotoxicity by amplifying NOX4 activity rather than its mRNA and protein expression levels. The results were quite unexpected but at the same time eye-opening for readers. There are some modifications before it could be accepted for publication.

1. In the second paragraph of the introduction, the authors mention that PM impairs neuronal integrity. It is recommended that the authors cite evidence which PM affects neuronal integrity or neuronal mitochondrial function in the following narrative, rather than focusing on pulmonary inflammatory responses.

2. It can only be seen from in Fig.4 J and L that the experiment was repeated three times. Please indicate the number of experimental repetitions in the figure note or in the Method section.

3. “Dose-dependent” means the magnitude or intensity of the effect increases or decreases as the dosage of the substance increases or decreases. Unlike the effects of PM10, PM2.5 can up-regulate the expression of VDAC1 while VDAC1 did not gradually increase with the increase of PM2.5 concentration. It cannot be called a dose-dependent relationship. This conclusion needs to be revised by the authors.

4. In the Discussion section, the authors conclude: This has elucidated the novel pathological role of PM in exacerbating neuronal cell death, with a focus on the upregulation of VDAC1 and subsequent mitochondria-derived ROS production stimulating NOX4 activity. This conclusion does not seem to be consistent with the results of the article, please revise it carefully.

5. Concentration units in micrograms per milliliter are expressed as “μg/mL” in the text and as “μg/ml” in the figures. The author is requested to unify the form of presentation.

Author Response

(The authors gave the same response as above.)
